# The Immunohistochemical Assessment of Neoangiogenesis Factors in Squamous Cell Carcinomas and Their Precursors in the Skin

**DOI:** 10.3390/jcm11154494

**Published:** 2022-08-02

**Authors:** Cloé Daneluzzi, Seyed Morteza Seyed Jafari, Robert Hunger, Simon Bossart

**Affiliations:** Department of Dermatology, Inselspital, Bern University Hospital, 3010 Bern, Switzerland; cloe.daneluzzi@students.unibe.ch (C.D.); morteza.jafari@insel.ch (S.M.S.J.); robert.hunger@insel.ch (R.H.)

**Keywords:** neoangiogenesis, vascular proliferation, cutaneous squamous cell carcinoma, cSCC, immunohistochemistry

## Abstract

Cutaneous squamous cell carcinoma (cSCC) is a common skin cancer. Well-limited forms can be easily treated with excision, but locally advanced cancers can, unfortunately, progress to metastasis. However, it is difficult to establish the prognosis for cutaneous squamous cell carcinoma and its potential to metastasize. Therefore, this study aimed to evaluate neoangiogenesis in cSCC, as it plays a major role in the dissemination of neoplasia. A literature review was performed on selected neoangiogenic factors (VEGF, ANG1/2, Notch1, CD31/34/105, EGF, etc.). Most of them, including VEGF, EGFR, and CD105, had more elevated levels in the advanced stages of the lesion. The same is true for Notch1, p53, and TGFβ, which are the most frequently mutated tumor suppressors in this type of skin cancer. In addition, the inhibition of some of these markers, using Ang1 analogs, inhibitors of EGFR, TRAF6, or combined inhibitors of EGFR and IGF-IR, may lead to a decrease in tumor size. In conclusion, this literature review identified diagnostic and prognostic markers, as well as possible factors that can be used for the targeted therapy of spinaliomas.

## 1. Introduction

The prevalence and incidence of cutaneous squamous cell carcinoma (cSCC) and other non-melanoma skin cancers vary across different regions of the world. The most important cause is the increased exposure of the skin to UV light through increased sun exposure and the use of solariums [1].

cSCC is one of the most frequent cancers in human and is more common in immunocompromised patients. Patients receiving immunosuppressive therapy for organ transplantation have a tenfold higher incidence of basal cell carcinoma (BCC), and the incidence of cutaneous squamous cell carcinoma (cSCC) is up to 65-fold higher. Both tumors can lead to death, especially spinalioma or squamous cell carcinoma, as the risk of metastasis is relatively high, in contrast to basal cell carcinoma, where the risk is virtually zero [2].

Unlike basal cell carcinomas, which usually arise from healthy skin with no previous lesions, cSCC usually arises from two precursors: actinic keratoses (AK) and Bowen’s disease (SCC in situ). Actinic keratoses are precancerous lesions with dysplastic keratinocytes caused by sun exposure. These early intraepithelial cSCCs are major risk factors for metastasis. Their growth can be self-limiting, or they can develop into invasive cSCC in approximately 1% of cases per year. In contrast, Bowen’s disease, which has dysplasia of all the layers of the epidermis, is already an intraepidermal carcinoma (carcinoma in situ) [3,4,5]. cSCC typically occurs on chronically sun-exposed skin areas such as the face, head, ear helix, back of the hands, and forearms. Chronic UV exposure is a risk factor. For approximately 1% of cSCCs, other risk factors, such as ionizing radiation, chemical noxae, and HPV infections, are necessary [5]. Metastatic disease is not common but is still possible. Metastasis is mostly lymphogenic in about 5% of patients, with a 5-year survival rate of approximately 25–50%. Metastases are more common in immunocompromised patients and patients with broader tumor extensions [1]. The therapeutic methods can be quite different depending on the tumor spread and histological features of the tumor. As an overview, the management of early or small cSCC primarily involves local therapies using various topical agents, such as immunological therapy (imiquimod) or local chemotherapy (5-fluorouracil), thermal ablation, surgical resection, or radiation. However, in locally advanced unresectable metastatic cases, palliative radiotherapy and/or classical chemotherapy may offer modest clinical benefits [6]. So far, there are no therapeutic standards for chemotherapy and immunotherapy. The response of patients with metastatic spinalioma to chemotherapeutic therapies is high; however, the treatment is not curative and is likely to cause recurrence. Recently, several new therapeutic strategies have emerged, such as EGFR inhibitors, tyrosine kinase inhibitors (gefitinib), and monoclonal antibodies against IgG1 (cetuximab). Immunotherapy with anti-PD1 antibodies (cemiplimab) has also been developed and is currently the therapy of choice for locally advanced or metastatic cutaneous squamous cell carcinoma in patients who are not candidates for surgery or radiotherapy [1,7]. 

In addition to conventional histopathology, immunohistochemical and molecular techniques are being used to determine possible new prognostic factors, including growth factors, neoangiogenesis, oncogenes, and tumor suppressor genes. Neoangiogenesis, or the growth of new blood vessels, is necessary for the development, spread, and dissemination of the tumor. Additionally, the vascular network is important for metastasis [8]. 

Thus, we performed a literature review on selected established neoangiogenic markers that may play a role in cSCC tumorigenesis.

## 2. Literature Search Strategy and Yielded Results

A systematic search of the PubMed electronic database was performed. All papers reporting an immunohistochemical analysis of selected angiogenic factors in cSCC were included (VEGF, VEGFR, angiopoietin, Tie, Notch1, CD31, CD34, CD105, EGF, EGFR, HIF, and TGFβ). The searches were not restricted by publication year, geographical situation, or language. Figure 1 illustrates the literature search and the results obtained. Articles on veterinary medicine, noncutaneous spinaliomas, and reviews were excluded. We identified 37 publications that reported angiogenesis in cutaneous squamous cell carcinoma in relation to the selected factors. The latest search date was July 2022. The database searches were supplemented by a reference search from the publications found in PubMed.

### 2.1. VEGF and VEGFR

Angiogenesis occurs early in several tumors and plays a major role in metastasis. Strieth et al. [9] analyzed tumor vascularization using CD31 in normal skin samples, AK, hypertrophic AKs, and early- and late-stage squamous cell carcinomas. Mean vascular density was similar in the normal dermis and AK and was only slightly increased in hypertrophic AKs and early-stage SCC (tumor thickness < 2 mm). Only late-stage cSCC that infiltrated the subcutis showed a significant increase in vascularization. Vessel density was independent of tumor location, degree of proliferation, or infiltration of inflammatory cells. Furthermore, tumor vascularization was not associated with VEGF expression. 

Bowden et al. [10] found that VEGF is expressed by blood vessel endothelial cells in the adjacent skin and tumor, as well as in the basal keratinocyte layer of the epidermis. VEGF expression was found in 32/41 cases of SCC and was significantly associated with the degree of tumor differentiation.

Elevated VEGFA levels are associated with several pathological conditions, including chronic inflammatory skin diseases and various types of skin cancers. VEGFA mRNA levels in SCC samples were approximately twice as high as those in healthy control samples according to the study by Kanitz et al. [11]. Data analysis by Tzoutzos et al. [12] showed that the molecular factors that control angiogenesis (including VEGF) are highly expressed in the tumors studied and that this measurement is positively correlated with tumor microvascular density. In addition, some VEGF receptors appeared to decrease when the tumor reached a certain size and achieved a microvascular network. Ciortea et al. [13] observed a correlation between VEGF expression and different stages of differentiation in cSCC. Among the well-differentiated (G1), moderately differentiated (G2), and poorly (G3) differentiated SCCs, most diagnosed G3 cases showed the deepest tissue infiltration and the highest VEGF positivity. Bălăşoiu et al. [14] also found that the immunoexpression of VEGF and EGFR was more pronounced in moderately or poorly differentiated forms than in well-differentiated SCC forms.

Nie et al. [15] found that the VEGF gene -460 C>T and -1154 G>A polymorphisms may serve as potential genetic markers for cSCC risk and prognosis. The TT and CT genotypes of the VEGF-460 C>T gene were significantly correlated with a decreased risk of cSCC, while the VEGF-1154 G>A gene and the AA genotype were significantly associated with reduced overall survival in cSCC patients.

The VEGF family and its corresponding receptors play critical roles in the regulation of angiogenic and lymphangiogenic processes. So far, VEGF-A, VEGF-B, VEGF-C, and VEGF-D are the most important members of the VEGF family. Studies have shown that VEGF-A exerts its effects through VEGFR-1 and -2. VEGF-B on VEGFR-1, VEGF-C, and VEGF-D affect VEGFR-2 and VEGFR-3 [8,16]. 

VEGF-A promotes angiogenesis during embryogenesis and in pathological conditions. VEGF-B maintains and allows the survival of pathologically formed blood vessels even under stressful conditions. VEGF-C and VEGF-D are important in lymphangiogenesis and lymphatic metastasis [8,16]. 

Mice exposed to UVB showed an increased presence of VEGF and CD31 markers compared with healthy skin. An injection of butoxamine, a selective β2 adrenergic receptor antagonist, induced an inhibition of angiogenesis, lower CD31 and VEGF expression than in the control group, and a decrease in the number of tumors in treated mice [17]. 

As in squamous cell carcinoma, these factors and their receptors have been detected in various organs and tumors [8]. Anti-VEGF therapies using antibodies against VEGFR receptors have already been used in ophthalmology. They are injected intravitreally to treat wet age-related macular degeneration or diabetic macular edema [18,19]. 

### 2.2. Angiopoietin 1,2 and Tie 2

Hawighorst et al. [20] studied the regulation of Ang1 and Ang2 mRNA expression in chemically induced mouse skin carcinogenesis. In normal vascularized skin, Ang1 is constitutively expressed by mesenchymal cells at low doses, while Ang2 mRNA is undetectable. The expression of Ang2 (without Ang1) strongly increased throughout the ongoing stages of mouse skin carcinogenesis. Strong Ang2 signaling was detected in the endothelial cells of tumor-associated vessels in cSCC. The proportion of tumor blood vessels covered by periendothelial smooth muscle cells also increased, indicating an enhanced vascular maturation in cSCC. The stable expression of Ang1, combined with a high rate of Tie2 receptor phosphorylation, resulted in the 70% inhibition of cSCC tumor growth. Therefore, the combined expression of Ang1/Tie2 has an inhibitory role in angiogenesis that occurs during tumor growth.

These findings suggest that a significant expression of Ang2 (antagonist of Ang1) may play a role in tumor progression and angiogenesis by blocking the stabilizing effect of Ang1 on Tie2.

Compared with healthy skin, the expression of Ang1, Ang2, and Tie2 is increased in HPV-associated warts [21]. No correlation was found between the number of markers and the duration of the disease or the number of warts. However, this study suggests the role of angiogenesis in the development of verruca vulgaris.

In cSCC, the role of HPV is not entirely clear. According to a review by Tampa et al. [22], HPV infection could be important in triggering carcinoma. The role of HPV in tumor progression is less important because the viral load is higher in precancerous lesions (such as actinic keratosis) than in established cSCC. However, chronic inflammation may also play a key role in carcinogenesis [22]. 

### 2.3. Notch1

Notch1 has been described as a tumor suppressor and oncogene in several studies [23,24,25,26,27]. The Notch signaling pathway can play opposing roles in different cancers depending on the cell and tissue context. This gene often acquires a loss-of-function mutation in cSCC. This finding supports the hypothesis of an oncosuppressive role of Notch1 in cutaneous carcinomas. 

Panelos et al. [23] investigated whether Notch1, like other Notch family proteins, is also down-regulated by UV radiation. They investigated the presence of Notch1 and its ligands in a series of premalignant and invasive skin carcinomas. Notch1 expression was decreased in actinic keratoses and in invasive cSCC that was localized to sun-exposed skin. Notch1 expression is variable, depending on the anatomic region affected and the tumor histology. While Notch1 is downregulated in UV-associated squamous photocarcinogenesis, sun-protected cSCC cells showed an upregulated Notch1 expression.

Wang et al. [24] identified Notch1 and Notch2 mutations in 75% of cSCC cases. The results show that Notch is the most prevalent tumor suppressor for these epithelial malignancies and that its loss of function plays a central role in disrupting microenvironment communication during cancer progression. The results of the study by Demehri et al. [26] suggested that tumor promotion in Notch1-deficient skin results from the additional contributions of fibroplasia, angiogenesis, and inflammation.

### 2.4. CD31, CD34, and CD105

The studies on CD31 are related to VEGF levels and are described in this paragraph. Florence et al. [28] aimed to demonstrate the role of angiogenesis in the progression of cSCC using an immunohistochemical study of endothelial markers. They quantified the microvascular area (Chalkley method) of actinic keratoses in superficial and invasive cSCC by comparing panendothelial (CD34) and neoangiogenic (CD105) markers at different times. The microvascular area in CD105-stained samples increased significantly with the progression of cSCC (from AK to cSCC). There were no differences between the different stages of the CD34 sections. All lesions showed significant increases in the microvascular area compared with the adjacent skin in both CD34- and CD105-stained samples. 

CD31 and CD34 were used as vascular endothelium markers, which are useful for estimating the vascular volume of the tissues. They are expressed in both normal and activated tumor vessels. CD105 is important for assessing neoangiogenesis and is highly expressed in activated endothelial cells. It has been shown that CD105-positive vessels are newly formed and fragile; hence, they can easily rupture or bleed [29]. 

The analysis of the vascular density of squamous cell carcinoma of the skin, using the immunohistochemical labeling of two main markers CD34 and smooth muscle actin (αSMA), shows a positive correlation between neovascularization and depth of tumor invasion. Furthermore, the lowest vascular density was found in G1-staged specimens and the highest values were in less differentiated tumors G3. The values were also higher than those measured in healthy skin samples [30]. 

Florence et al. compared p53 immunoexpression at different stages of cSCC with neovascularization (CD105 expression) and cell proliferation (Ki67 expression) [31]. They studied p53- and Ki67-immunolabelled cells in three groups: AK, superficial cSCC, and superficial-invasive cSCC. Neoangiogenesis (CD105) was quantified in each group using the Chalkley method. The study showed no significant differences in the number of p53- and Ki67-positive cells among the three groups. Significant positive correlations were found between the microvascular CD105 area and the number of p53-positive cells in superficial invasive cSCC and between the frequency of p53- and Ki67-positive cells in invasive cSCC. The immunoexpression of p53 and Ki67 did not increase with the progression of cSCC. Neovascularization is associated with p53 expression in the initial phase of invasion and proliferative activity in the invasive stages of cSCC. Loss of the p53 tumor suppressor function may directly contribute to skin carcinogenesis. 

### 2.5. EGF and EGFR

The activation of EGFR promotes proliferation, migration, adhesion, differentiation, and cell survival; similarly, the activation of IGF-IR enhances proliferation; protects from apoptosis; and stimulates tissue migration, invasion, and angiogenesis. Galer et al. [32] showed that IGF-IR and EGFR expression levels are consistently and simultaneously increased in cSCC cell lines. They used a combination of the A12 (anti-IGF-IR) antibody and cetuximab (anti-EGFR) to simultaneously block EGFR and IGF-IR activation, directly blocking the tumorigenic and angiogenic effects of these receptors, resulting in a significant reduction of tumor volume. In a cSCC mouse model, dual inhibition with the A12 antibody and cetuximab reduced the tumor volume by 92%, with increased mouse survival. In comparison, when the two agents were applied separately, the tumor volume only decreased by 50%. 

As mentioned above, Bălăşoiu et al. [14] studied the immunolabeling of VEGF and EGFR in cSCC. EGFR was present in 85% of squamous cell carcinoma cases, labeling 15–85% of cells. It is present in 66.6% of well-differentiated squamous cell carcinomas, labeling 10–30% of cells with a weak or moderate staining intensity. In moderately differentiated squamous cell carcinomas, EGFR immunostaining occurred in 85.7% of cases, moderately or strongly labeling 25–60% of the tumor cells. Finally, an EGFR immunohistochemical response was present in all poorly differentiated squamous cell carcinoma cases, moderately or strongly staining 45–85% of the tumor cells. Analysis of the staining percentages showed significantly higher values for moderately or poorly differentiated squamous cell carcinoma than for well-differentiated cSCC. 

Dysregulation of EGFR is closely associated with tumorigenesis and has been observed in several types of tumors. According to Zhang et al. [33], EGFR is overexpressed in cSCC cells, especially in the advanced or metastatic stages. Genetic analysis showed that EGFR has a very low mutation frequency, similar to RAS, whose mutations are rarely observed in cSCC. Owing to a dysregulated EGFR activation in the absence of EGFR or RAS mutations, targeting EGFR is an important therapeutic strategy in cSCC. Cetuximab is already used in patients who cannot undergo surgery or radiotherapy [34]. TRAF6 promotes oncogenesis by inhibiting apoptosis and stimulating the proliferation and invasion of cancer cells. However, the role of TRAF6 in cSCC remains unclear. Zhang et al. [33] found that TRAF6 is required for EGF-induced cell transformation. It plays a crucial role in cSCC growth and metastasis via the EGF associated signaling pathway. The growth of cSCC cells was blocked in a TRAF6-knockdown cell line, resulting in smaller tumor sizes and delayed tumor formation. On histopathological examination, the tumors showed decreased Ki-67 and EGFR expression compared with the control group.

### 2.6. Other Factors

Cammareri et al. [35] investigated whether the loss of TGFβ signaling is a common event in sporadic cSCC. Using a percentage variance criterion of >10%, they detected TGFβR1 and TGFβR2 mutations in 22% and 30% of the primary cSCC samples, respectively. Overall, mutations in the TGFβ receptors occurred in 43% of the primary cSCC samples. These mutation events were exceeded in frequency only by mutations in Notch1/2 (86%) and TP53 (63%). In sporadic cSCC, the oncogenic activation of RAS occurs in only 9% of samples. TGFβ receptor mutations were not identified in either distant or peri-lesional skin. These results suggest that the inactivation of TGFβ signaling may be an initiating event in sporadic cSCC. 

If collagen VII levels are lower than the norm, as may be the case in certain genetic diseases such as recessive dystrophic epidermolysis bullosa (RDBE), the occurrence of aggressive cSCC is more frequent. A lack of collagen VII induces an increased TGFβ expression that in turn increases VEGF levels and angiogenesis. The intradermal injection of collagen VII has already shown results in mice with RDEB that had correction of blistering with the incorporation of collagen VII into the basement membrane. In the clinic, the use of collagen for its anti-angiogenic utility could be considered [36]. 

HIF-1α is elevated in cSCC, which upregulates p21 expression [37], and p21 inhibits the growth of keratinocytes and influences their differentiation in response to various stimuli. An HIF-1α-inhibiting RNA injection in rats was also found to decrease the amount of p21 and thereby induce skin hyperplasia [38]. 

Seleit et al. [39] immunohistochemically stained HIF-1α in cSCC skin biopsies and normal skin samples. In patients with cSCC, HIF-1α was expressed in 100% of the tumor areas, while only 90% of their healthy skin biopsies were positive. There was a stronger staining signal in the tumor cells than in normal cells; furthermore, staining was diffuse in the nuclei and cytoplasms of the tumor cells.

Two different studies [40,41] indicated that SOX18, a transcription factor involved in endothelial cell differentiation, was upregulated in skin cancers whereas it was nearly absent in healthy skin samples. Neinaa et al. [40] showed that the level of SOX18 was higher in cSSC lesions with a significant difference from actinic keratoses. In addition, the more severe forms of the tumors also showed higher levels of this marker. The vascularity was also studied, and a positive correlation was found with the intensity of SOX 18 measured in the different samples. 

CYLD is an already recognized tumor suppressor in several different tumors. Alameda et al. found that transgenic mice without functional CYLD showed activation of the NF-κB pathway, resulting in chronic inflammation and the spontaneous appearance of CSC. They then analyzed its action at the skin level in transgenic mice expressing the wild type form of CYLD. They found a decreased activation of the nuclear factor kappa B (NF-κB) pathway, which promotes epithelial differentiation and inhibits proliferation. The results also show a decrease in the proliferation and numbers of skin tumors as well as a decrease in inflammation and angiogenesis. The vessels found are also more mature and impermeable. When exposed to various stresses (the mitogen TPA), CYLD-expressing mice showed less hyperplasia than control mice [42]. 

In the same way, Podoplanin expression was also upregulated in samples of malignant tumors such as SCC compared with healthy skin. Lymphatic vascular density was significantly positively correlated with Podoplanin value in SCC [43]. 

The transmembrane protein LRIG2 is involved in the feedback loop for the regulation of ERBB receptors. Hoesl’s [44] analysis of transgenic mice overexpressing LRIG2 showed protein changes in the EGF/ERBB system as well as increased tumor progression with a more rapid development of cSCC-like phenotype. The analysis of human cSCC samples also showed increased levels of LRIG2 compared with a healthy skin sample. 

From a diagnostic point of view, autofluorescence could allow the distinction between benign and malignant lesions and give an indication of the resection margin. Measurements have shown a decrease in autofluorescence (after irradiation with a wavelength of 400–430 nm) in cancerous lesions such as BCC and cSCC, especially in relation to histological alterations such as elastosis, fibrosis, severe hyperkeratosis, and angiogenesis. However, cell atypia did not play a significant role in the difference in autofluorescence emission [45]. 

## 3. Discussion

Different markers have been identified to date, but the most important factor is the VEGF family. VEGF-A, in particular, is important because it promotes angiogenesis in pathological conditions. Targeted therapy using antibodies against VEGFA has existed for decades and is mainly used in ophthalmology [18,19]. 

Notch 1 has previously been described as an oncogene or a tumor suppressor depending on the tumor type. In cSCC and other cutaneous carcinomas, this marker acts as a suppressor. Notch1 or 2 mutations have been identified in 75% of cSCC cases. Therefore, it is the most common tumor suppressor in cSCC. Similarly, TGFβ is one of the three most commonly mutated tumor suppressors in cSCC, along with Notch1 and p53 [24]. 

EGF and EGFR have been extensively studied. In addition to EGFR antibodies, several therapies using antibodies that abort EGFR signaling are available on the market [1,34]. 

The relative gene expression levels of Ang1, Ang2, and Tie2 were higher in HPV-associated warts compared with normal skin, suggesting that the upregulation of these biomarkers may play a role in the development of such warts [21]. HPV infection could promote cSCC, especially in the context of chronic inflammation, such as actinic keratosis [22]. 

CD31 and CD34 are established markers for assessing the vascular endothelium. Furthermore, CD105 has an affinity for active cells in the vascular endothelium. Therefore, it has been described as a suitable neoangiogenic marker [28,29]. 

Since VEGF is highly expressed in advanced tumors, the correlation between poorly differentiated tumors and high VEGF levels could be a starting point for prognostic marker research. Poor differentiation is associated with deeper tissue infiltration; therefore, there is a high risk of metastasis. Measuring VEGF levels could help estimate the likelihood of metastatic risk [10,13,14]. The role of β2 adrenergic receptor antagonist in reducing the development of UVB-induced cancer lesions, needs to be investigated as it may represent a new therapeutic opportunity [17]. 

VEGF can also be used as a genetic marker because the AA genotype of the VEGF-1154 G>A gene is associated with a limited lifespan. It is important to note that unlike CD31/34, VEGF does not provide information about tumor vessel density [15]. 

This marker should be measured in the blood vessels located in the tumor and surrounding skin to provide additional information about the degree of differentiation and, thus, the possible infiltration of carcinoma into the tissue [10]. 

It would be interesting to use Ang1 and Ang2 for diagnostic and therapeutic purposes based on the observations of these biomarkers. Ang2 is not usually expressed in healthy tissues but is found in the endothelial cells of tumor vessels. Therefore, it could serve as a diagnostic marker in skin carcinogenesis. In addition, Ang1 could be a therapeutic intervention marker, as it inhibits the growth of cSCCs by phosphorylating the Tie2 receptor [20,21]. 

Since Notch1 is decreased in AK and spinaliomas located in sun-exposed areas, it may have some diagnostic role in different types of skin diseases (sun-exposed vs. unexposed). Notch1 is known to be the most important tumor suppressor in cSCC and thus could serve as a prognostic and diagnostic marker [23]. 

CD31 expression increases with the infiltration stage of the tumor. Therefore, CD31 would be more useful as a prognostic marker than CD34. Since CD31 and CD34 only indicate the number of vessels present, it would be more useful to use the CD105 marker since it provides information about the dynamic changes in the vessels. Thus, it is possible to identify the density of new vessels. It could be used as a prognostic marker because a progressive increase in CD105 levels is linked to the progression of cSCC. There is also a correlation between CD105 levels and p53 mutations in advanced invasive stages. P53 is known to have a tumor-suppressive function in cSCC. Therefore, the amount of CD105 may indicate the mutation risk of p53 [31]. 

EGFR could be used as a prognostic marker since it is elevated in poorly differentiated spinaliomas. No other mutations have been shown to occur with EGFR mutations. Therefore, targeted therapy against EGFR or signaling may be effective. These possibilities have already been researched, and several new therapies have recently emerged. Another important point is the essential presence of TRAF6, which when knocked down induces a degradation of EGFR resulting in downregulating EGFR signaling. The inhibition of TRAF6, while decreasing EGFR and other signaling pathway, halts tumor cell growth and therefore has a huge therapeutic potential. Another therapeutic possibility could be addressed by the combined inhibition of EGFR and IGF-IR, which could lead to a reduction in tumor volume [1,32,33]. 

The HIF-1α biomarker is increased in tumor cells and could be used as a prognostic marker because it affects the p21 protein. P21 is upregulated in the presence of HIF-1α, thereby inhibiting the growth of keratinocytes and promoting their differentiation when triggered by various stimuli. The inhibition of HIF-1α could be a possible therapeutic strategy to stop the differentiation and allow for the continued growth of healthy keratinocytes [37,38]. 

The results found for SOX18 could indicate a possible role for this transcription factor in the malignancy and aggressiveness of the tumor, given that it is not found in healthy skin and is found in lesser quantities in benign tumors such as actinic keratoses than in malignant tumors such as cSCC. Furthermore, the vessel density was higher in samples with more intensity of SOX18 [41]. 

Finally, it is important to reiterate the three most commonly mutated tumor suppressor genes in cSCC: Notch1/2, p53, and TGFβ in 86%, 63%, and 43% of the cases, respectively. Corresponding mutations in the three tumor suppressor genes are found in advanced/invasive cSCC. Therefore, it would be advisable to study and identify these three tumor suppressors and their possible mutations to obtain a more accurate idea of the possible course of the tumor [35]. 

In summary, some markers were found to have potential diagnostic, prognostic, and therapeutic values in cSCC:A potential diagnostic role was found for Ang2, which is expressed only by the endothelium of tumor vessels but is probably not specific for SCC, and Notch1, which is decreased in sun-exposed areas. Autofluorescence should also be considered in the future as a noninvasive diagnostic tool complementary to the current gold standard.VEGF, Notch1, CD105, p53, EGFR, HIF-1α, p21, and TGFβ could be prognostically important factors. All of these factors are highly elevated in the advanced stages of cSCC.Possible therapeutic strategies include the use of β2 adrenergic receptor antagonist, Ang1 analogs, anti-EGFR, anti-TRAF6, anti-EGFR combined with anti-IGF-IR, anti-HIF-1α, and collagen VII.

All these suggestions require further research to investigate the potential of these factors as diagnostic, prognostic, and therapeutic markers for cSCC, and to test their validity for possible targeted therapy.

## 4. Limitations

As squamous cell carcinomas are localized in most cases, excision usually provides a cure, although a small percentage of these lesions can become invasive and lead to death. For this reason, it was difficult to find articles that focused on various prognostic factors for this particular tumor. Since there are only a few relatively new studies on these new factors, currently, there are only a few published concrete results in the form of clinical trials.

## 5. Conclusions

Several factors show interesting behaviors during cSCC carcinogenesis; some could provide important information about the probable course of the tumor, while others could be very important elements for targeted therapy. Therefore, it is necessary to further develop the different results of this study to improve the treatment modalities for patients with squamous cell carcinoma.

## Figures and Tables

**Figure 1 jcm-11-04494-f001:**
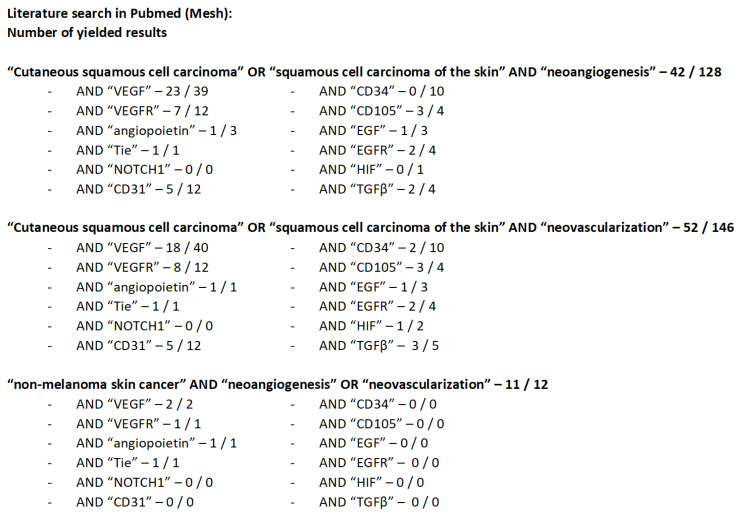
The search terms and the yielded results.

## Data Availability

Data is contained within the article.

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
