# Peer review of "The Immunohistochemical Assessment of Neoangiogenesis Factors in Squamous Cell Carcinomas and Their Precursors in the Skin"

_jcm, 2022, doi:10.3390/jcm11154494_

Round 1

Reviewer 1 Report

This review discusses a relatively limited number of studies: excluding an inappropriately formatted reference 6, only 35 references are included in the review. Moreover, 43% (15 out of 35) of the papers discussed in this review were published between 1998 and 2012, and only four were published in the last three years. Therefore, the enthusiasm for this work is diminished due to a limited number of recent publications reviewed. 

As shown in Figure 1, which delineates the literature search strategy, the authors employed the terms “cutaneous squamous cell carcinoma” and “neoangiogenesis,” plus various specific molecules implicated in angiogenesis, to obtain the list of the publications included in the review. The authors should consider additionally including the related terms for their search, such as “skin squamous cell carcinoma,” “non-melanoma skin cancer,” “neovascularization,” “angiogenesis,” etc. A brief Pubmed search using these additional terms has yielded several relevant studies overlooked by the authors, including, but not limited to the following publications:

·       PMID: 32662888 (SOX18), 2020

·       PMID: 31688008 (Podoplanin), 2020

·       PMID: 31580518 (LRIG2), 2019

·       PMID: 28593886 (The increased vascularization at the invasive front of CSCC), 2017

·       PMID: 26476432 (Type VII collagen suppresses TGFbeta signaling and angiogenesis in CSCC), 2015

The authors should also consider searching the literature for novel potential regulators of neoangiogenesis in cSCC, such as epigenetic regulators, non-coding RNAs, etc. 

The following statement needs to be revised (lines 268-269):

“The Ang and Tie biomarkers have been studied in HPV-associated warts; thus, angiogenesis may be important for the development of such warts,” for example, as follows: “The relative gene expression levels of Ang1, Ang1, and Tie2 were higher in HPV-associated warts compared with normal skin, suggesting that upregulation of these biomarkers may play a role in the development of such warts.”

The following statement also needs to be revised (line 272):

“CD31/CD34 is an established marker for assessing the vascular endothelium.” It is incorrect, as CD31 (PECAM1) and CD34 are encoded by two different genes located at different chromosomes, so these molecules are two separate markers rather than one single marker.

The following statement is incorrect and needs to be revised (lines 308-310):

“Another important point is the essential presence of TRAF6, which induces mutations in EGF-modulated cells”. While the study referenced as # 32 indeed shows the essential role of TRAF6 in cSCC malignant phenotypes, contrary to the statement mentioned above, there is no evidence in that report that TRAF6 “induces mutations in EGF-modulated cells.”  In fact, TRAF6 knockdown promotes the degradation of EGFR and thus downregulates EGFR signaling. As detailed in the report, several other signaling pathways known to be essential for cSCC development are also inhibited due to TRAF6 knockdown (Ref. #32).

Reference # 6 is unacceptable and needs to be eliminated or replaced.

The term “under-regulated” (lines 162, 167) should be replaced with the commonly used term “down-regulated.”

The authors state that “targeting EGFR could become a therapeutic strategy in cSCC” (line 229). It should be pointed out that EGFR inhibitor- (i.e., cetuximab)–based therapies have been tested for cSCCs in clinical trials (see, for example, PMID: 21810686, PMID: 32064041). Of note, as discussed in PMID: 21810686, tumor EGRF expression levels were not associated with treatment efficacy.

Author Response

please see attachment Cover Letter Reviewer No.1 on page 1-2. 

Reviewer 2 Report

Daneluzzi and co-authors presented a manuscript entitled: “Immunohistochemical assessment of neoangiogenesis factors in squamous cell carcinomas and their precursors in the skin”. They evaluate several papers reporting an immunohistochemical analysis of neoangiogenetic markers in cutaneous squamous cell carcinoma (cSCC) trying to identify possible diagnostic and prognostic markers and new pharmaceutical targets.

The topic presented is interesting and well presented. However, their focus is a review on the evaluation of neoangiogenic markers in cutaneous squamous cell carcinoma (cSCC). Why do the authors generalize by using the term non-melanoma skin cancer? (Please see abstract and introduction) it may be confusing for the readers cause the term non-melanoma skin cancer is related to all skin neoplasms that are not melanoma. They should focalize their description on cSCC to avoid misunderstanding. 

Author Response

Please see attachment Cover Letter Reviewer No.2 on page 3. 

Round 2

Reviewer 1 Report

The revised manuscript is significantly improved.

Minor comment:

Line 324: instead of "knockdowned", should be "knocked down"